# Effect of Plant Growth Regulators on Osmotic Regulatory Substances and Antioxidant Enzyme Activity of *Nitraria tangutorum*

**DOI:** 10.3390/plants11192559

**Published:** 2022-09-28

**Authors:** Dom Alizet Didi, Shiping Su, Faisal Eudes Sam, Richard John Tiika, Xu Zhang

**Affiliations:** 1College of Forestry, Gansu Agricultural University, Lanzhou 730070, China; 2College of Enology, Northwest A&F University, Yangling 712100, China

**Keywords:** plant growth regulator, abscisic acid, indole acetic acid, gibberellic acid, osmotic regulatory substances, antioxidant enzymes, *Nitraria tangutorum* Bobr.

## Abstract

Plant growth regulators (PGRs) are natural hormones and synthetic hormone analogues. At low concentrations, PGRs have the ability to influence cell division, cell expansion, and cell structure and function, in addition to mediating environmental stress. In this study, experiments were conducted to determine how exogenous PGRs indole acetic acid (IAA), abscisic acid (ABA), and gibberellic acid (GA) influenced osmotic regulatory substances and activity of antioxidant enzymes in *Nitraria tangutorum*. Using a completely randomized design, IAA, ABA, and GA3 were applied as foliar spray at concentrations of 50 mg/L, 100 mg/L, 150 mg/L, and 200 mg/L to *N. tangutorum* shrubs. Some selected shrubs did not receive any treatment and served as the control (Ck). The results showed that the foliar spray of IAA, ABA, and GA3 significantly increased the content of osmotic regulatory substances (soluble sugar, soluble protein, and proline) and antioxidant enzymes (SOD and POD) at most concentrations. In addition, the malondialdehyde (MDA) content significantly reduced after treatment, but after regrowth of coppiced shrubs, lipid peroxidation increased and was still lower than Ck. Our study provides evidence that 100 mg/L 150 mg/L, and 200 mg/L concentrations of IAA, ABA, and GA3 treatments are effective for enhancing osmotic regulatory substances and the activity of antioxidant enzymes in *N. tangutorum*, which offers an effective strategy not only for increasing tolerance to abiotic and biotic stresses, but also improving the adaptability of *N. tangutorum* shrubs to the environment.

## 1. Introduction

Plant growth regulators (PGRs) were first discovered in plants in the 20th century. PGRs are a variety of diverse molecules known as plant hormones that act as chemical messengers to control plant growth and development [1,2]. PGRs exist in both natural (extracted) and synthetic (manufactured) forms. Although a synthetically produced plant growth regulator is structurally identical to a plant regulator or hormone, it is not considered to be one [2]. The production and growth of roots, shoots, buds, flowers, and fruits can be controlled by the use of synthetic PGRs [3]. PGRs are compounds that promote plant growth and increase their stress tolerance under various stress conditions [4,5]. PGRs affect a variety of plant traits, including plant height, number of leaves, leaf area index, dry matter, chlorophyll concentration, and photosynthetic parameters, among others [6]. PGRs can also improve plant tolerance to various abiotic stimuli, increase antioxidant capacity, and accelerate plant growth [7]. PGRs have been synthesized and can be exogenously applied to plants to control their growth and development. PGRs come in a variety of forms, including auxins such as indoleacetic acid (IAA), abscisic acid (ABA), and gibberellic acid (GA). 

In plants, IAA is synthesized by tryptophan or tryptophan-independent mechanisms [8]. IAA has been found to promote plant growth [9]. It is a tiny molecule with a simple structure that helps in the growth and development of numerous plant organs [10]. It can also help with more than just seed germination [11]. In particular, IAA can increase sugar accumulation (glucose, fructose, and total soluble sugars), stomatal conductance, photosynthetic rate, and pigment levels in plants [12,13]. In addition, IAA can increase the density of leaf veins, leading to an enhanced photosynthetic capacity of leaves [14]. IAA foliar spray was reported to improve yield, biochemical properties, and antioxidant activity [15]. Plants require IAA not only for development but also for producing antioxidant enzymes and bioactive chemicals [15]. Anam et al. [16] reported that exogenous IAA application increases the activity of antioxidant enzymes in carrots. 

Abscisic acid (ABA) also controls a variety of plant processes and is the major regulator of abiotic stress resistance in plants [17,18]. ABA has many crucial functions in plants at molecular and physiological levels, especially under environmental stress [19,20]. This phytohormone acts as a messenger in response to abiotic stress and stimulates the expression of genes encoding antioxidant enzymes, which in turn increases the activity of these enzymes [21,22]. Reportedly, ABA controls photosynthesis and leaf transpiration, stimulates stomatal closure, limits growth of the aerial component, controls dormancy, and influences plant response to stress [23].

GA3 is another important plant signaling hormone that promotes several physiological and developmental processes in plants, including seed germination, flowering, cell division and maturation, and root formation. GA3 can also increase plant tolerance to environmental stresses such as salt, cold, drought, and trace element stress [24,25]. Furthermore, to mitigate the drastic effects of environmental stress, GA3 increases antioxidant capacity and osmoprotectants and minimizes lipid peroxidation [26]. Sakya et al. [27] also reported that GA3 could increase and regulate the ability of antioxidants. To help plants withstand challenging environmental conditions, osmolytes and osmoprotectants have long been recognized as effective methods to resist abiotic stress [28,29,30,31]. Sharma et al. [32] reported that osmolytes are osmoprotective solutes that are both organic (like glycine betaine, proline, carbohydrates, and proteins) and inorganic (such as Ca^2+^, K^+^, PO_4_^3−^, NO^3−^, and SO_4_^2−^), and help cells retain water without interfering with regular metabolism. Moreover, the antioxidant system of plants consists of both enzymatic and nonenzymatic antioxidants that help the plant survive under stressful conditions [33].

As desertification has become a reference in China, the restoration, propagation, and protection of desert shrubs that can withstand the sand of desert grasslands are very important in combating desertification. *Nitraria tangutorum* Bobr. of the *Zygophyllaceae* family is native to China [34]. It is a tiny, distinctive, and widespread sand-fixing shrub in Inner Mongolia and China’s arid desert regions [35]. *N. tangutorum* is an important component of desert vegetation and is resistant to a variety of stresses, including wind erosion [36], sand burial [37], drought [38], and salt and alkali stress [39,40]. This resistance is related to its well-developed root system, small and fleshy leaves, and quickly grown branches. Therefore, *N. tangutorum* is essential for maintaining the diversity of flora in desert regions, preventing wind erosion, fixing sand, and improving the physical and chemical properties of the soil. In addition, *N. tangutorum* provides a significant source of income for the local populace, as its fruits, known as “desert cherries”, are used to make beverages and medicines [41], and their waste (e.g., dry twigs and fallen branches) is often used as firewood by local residents [42]. Regarding this shrub, several studies have been conducted only in relation to different stress situations (salt stress, drought stress, water stress); however, no studies have been conducted on the effect of PGRs on the metabolism of *N. tangutorum*, particularly the impact on osmotic regulatory substances and antioxidant enzyme activity. Therefore, this work aimed to investigate the effect of three PGRs (IAA, ABA, and GA3) on osmotic regulatory substances and antioxidant enzyme activity in *N. tangutorum* after foliar sprays and before and after coppicing. 

## 2. Results

### 2.1. Effect of IAA on N. tangutorum Osmotic Regulatory Substances 

It is evident from (Figure 1A) that the soluble sugar (SS) content was improved compared with the control (Ck) by the foliar application of IAA at various concentrations. The SS content increased in shrubs treated with 50 mg/L, 100 mg/L, 150 mg/L, and 200 mg/L of IAA by almost 1.57 times, 1.92 times, 2.43 times, and 1.94 times, respectively. After regrowth, only 50 mg/L concentration significantly increased the SS content by 91%, whereby IAA treatment at 100 mg/L had no significant effect compared with Ck. In contrast, IAA treatment at 150 mg/L and 200 mg/L significantly decreased the SS content by 74% and 78%, respectively, compared to Ck.

Foliar spray of IAA at different concentration levels significantly elevated the soluble protein (SP) content in *N. tangutorum* shrubs (Figure 1B). The results showed that all the concentration levels of IAA increased the content of the SP compared to the control after treatment and after regrowth in the order 200 mg/L > 150 mg/L > 100 mg/L > 50 mg/L > Ck. Among the treatment levels, the highest concentration of SP was observed at an IAA concentration of 200 mg/L after treatment (7.7 mg/Fresh weight (Fw)) and after regrowth (8.9 mg/Fw) compared with Ck (3.7 mg/Fw and 2.0 mg/Fw, respectively).

Similar to the results obtained for SP, foliar spray of IAA at concentrations of 50 mg/L, 100 mg/L, 150 mg/L, and 200 mg/L also significantly (*p* < 0.05) increased the content of proline (PRO) after treatment by 1.67 times, 2.42 times, 2.72 times, and 3.24 times, respectively, compared with the control. Furthermore, an increase in PRO content was observed in 50 mg/L, 100 mg/L, 150 mg/L, and 200 mg/L treated shrubs after regrowth by 1.49 times, 2.08 times, 2.24 times, and 2.42 times, respectively. Furthermore, IAA application at 200 mg/L resulted in the highest content of PRO both after IAA application (6.20 mg/Fw) and after regrowth (4.46 mg/Fw) of the plant compared with the other concentration levels and Ck (Figure 1C).

### 2.2. Effect of IAA on Antioxidant Enzyme Activity in N. tangutorum

The superoxide dismutase (SOD) activity increased significantly in *N. tangutorum* after IAA treatment at all concentrations (50 mg/L, 100 mg/L, 150 mg/L, and 200 mg/L) and after regrowth (except 50 and 100 mg/L) compared to the control (Figure 1D). After IAA treatment, the highest SOD activity was observed in plants treated with 200 mg/L IAA (69.44%) compared with Ck. Among all the treatments, the SOD activity was higher by 87.63% in the leaves of plants treated with IAA at 150 mg/L compared with the control after regrowth (Figure 1D).

Peroxidase (POD) activity in *N. tangutorum* shrubs treated with IAA is shown (Figure 1E). After IAA treatment at concentrations of 100 mg/L, 150 mg/L, and 200 mg/L, the POD activity significantly increased compared with Ck by 44.11%, 66.12%, and 34.19%, respectively. After regrowth, two concentration levels, 150 mg/L and 200 mg/L, significantly increased the POD activity by 65.36% and 36.66%, respectively, compared with the control. Moreover, among the treatments, 150 mg/L treatment was the most important as it caused the highest POD activity in *N. tangutorum* both after treatment (0.35 mg/Fw) and after the regrowth (0.45 mg/Fw) of the shrubs compared to other treatments and the control (Figure 1E).

### 2.3. Effect of IAA on Malondialdehyde Content in N. tangutorum

After IAA application at different concentrations, the results showed that the different concentration levels of IAA significantly decreased the malondialdehyde (MDA) content compared with Ck, especially at 200 mg/L (Figure 1F). After regrowth, plants treated with 150 mg/L of IAA had the highest concentration among the other treatments, while 200 mg/L treated shrubs had the lowest concentration than the control. In particular, the MDA content in shrubs treated with 150 mg/L of IAA increased almost 0.44 times compared to other concentrations before coppicing, but the increase was not significant.

### 2.4. Effect of ABA on N. tangutorum Osmotic Regulatory Substances 

The soluble sugar (SS) content of *N. tangutorum* was significantly different (*p* < 0.05) at various concentrations of ABA applied. The SS content of shrubs treated with ABA at concentrations of 100 mg/L, 150 mg/L, and 200 mg/L significantly increased compared with Ck (Figure 2A). However, at 50 mg/L concentration, no significant difference was observed compared with Ck. After regrowth, ABA significantly increased the content of SS more than Ck, particularly at 50 mg/L, 100 mg/L, and 200 mg/L. After ABA treatment, 200 mg/L was more efficient among the test samples as it increased the SS content averagely, by 1.37 times compared with the others.

The soluble protein (SP) content of all the treatments is shown in Figure 2B. After foliar spray of ABA, 150 mg/L recorded significantly (*p* < 0.05) higher SP contents, about 1.33 times, 1.17 times, 1.23 times, and 1.38 times higher than that of Ck, 50 mg/L, 100 mg/L, and 200 mg/L, respectively. Similar results were found after the regrowth of *N. tangutorum* shrubs, whereby ABA treatment at 150 mg/L still recorded the highest content among all the treatments and 200 mg/L recorded the lowest (Figure 2B).

As shown in Figure 2C, the proline (PRO) content in *N. tangutorum* shrubs was significantly higher by 1.5-fold, 1.7-fold, 2.0-fold, and 2.8-fold at concentrations of 50 mg/L, 100 mg/L, 150 mg/L, and 200 mg/L, respectively, than the control after treatment. After regrowth, the PRO content of the shrubs treated with ABA at all concentrations (except 50 mg/L) was also significantly higher than Ck (Figure 2C). Thus, the PRO content significantly increased by 2.7 times, 3.8 times, and 4.2 times in 100 mg/L, 150 mg/L, and 200 mg/L, respectively, compared with the control. Among the treatment levels of ABA, 200 mg/L concentration caused a higher increase in PRO content after treatment and regrowth of the shrubs.

### 2.5. Effect of ABA on Antioxidant Enzyme Activity in N. tangutorum

The treatment of *N. tangutorum* shrubs with ABA at different concentration levels (except 150 mg/L) significantly increased the superoxide dismutase (SOD) activity averagely by 22.4% compared to the control. After regrowth, the SOD activity of plants treated with 50 mg/L and 100 mg/L significantly increased by 31.8% and 15.6%, respectively, compared to the control (Figure 2D). Furthermore, 50 mg/L concentration recorded the highest SOD activity (40.0 mg/Fresh weight), whereas 200 mg/L recorded the least (22.8 mg/Fw) compared to Ck after regrowth.

An increase in peroxidase (POD) activity was also observed after ABA treatment, especially at concentrations of 50 mg/L, 150 mg/L, and 200 mg/L (Figure 2E). Among the treatments, POD activity in plants treated with 200 mg/L ABA was averagely 2.57 times higher than those of Ck and other concentration levels. Thus, the POD activity increased in the treatments in the order 200 mg/L > 150 mg/L > 50 mg/L > Ck > 100 mg/L (Figure 2E). After regrowth, 50 mg/L, 100 mg/L, and 150 mg/L concentrations had no significant effect on the POD activity compared with the Ck. However, at 200 mg/L, the highest POD activity (0.60 mg/Fw) was observed. The POD activity in the treatments after regrowth increased in the order 200 mg/L > 100 mg/L > 50 mg/L > Ck > 150 mg/L (Figure 2E).

### 2.6. Effect of ABA on Malondialdehyde Content in N. tangutorum

Malondialdehyde (MDA) activity significantly decreased (*p* > 0.05) after treatment with ABA at all concentrations compared with Ck (Figure 2F). ABA treatment at 200 mg/L recorded the lowest amount of MDA than other concentrations and Ck. After regrowth, the MDA content of all concentration levels of ABA increased, which was not significant prior to coppicing but was lower than the CK. Compared to the control, the MDA content in shrubs treated with 150 mg/L ABA was the most important as it increased by 0.27-fold compared with other concentrations prior to coppicing. In contrast, the MDA content in shrubs treated with 200 mg/L ABA was the lowest (decreased by 0.2 times) compared to CK after regrowth (Figure 2F).

### 2.7. Effect of GA3 on N. tangutorum Osmotic Regulatory Substances

Foliar application of GA3 at concentrations of 50 mg/L, 100 mg/L, 150 mg/L, and 200 mg/L significantly increased the soluble sugar (SS) content in *N. tangutorum* shrubs compared with Ck by 1.27 times, 1.48 times, 1.72 times, and 1.79 times, respectively (Figure 3A). After the regrowth of *N. tangutorum* shrubs, an increase in SS content was observed in all shrubs treated with GA3, regardless of the concentration level. In particular, the content of shrubs treated with 50 mg/L, 100 mg/L, 150 mg/L, and 200 mg/L of GA3 increased substantially by 2.56-fold, 3.53-fold, 2.90-fold, and 3.0-fold, respectively, compared to Ck.

Figure 3B reveals that after applying GA3 treatment at 100 mg/L, the soluble protein (SP) content in the *N. tangutorum* shrubs was not significant (*p* > 0.05) compared with CK. The SP content in *N. tangutorum* shrubs treated with GA3 before coppicing was significantly (p < 0.05) higher at concentrations of 50 mg/L, 150 mg/L and 200 mg/L by 1.49-fold, 1.94-fold, and 2.13-fold, respectively, compared with Ck. After regrowth, the SP content in shrubs treated with 50 mg/L, 100 mg/L, 150 mg/L and 200 mg/L GA3 further increased significantlyby 5.04-fold, 6.42-fold, 7.08-fold, and 7.80-fold compared with Ck. Among the test samples, the highest content of SP was recorded in shrubs treated with 200 mg/L GA3 after regrowth (2.13 mg/Fw).

The results of proline (PRO) content in the *N. tangutorum* shrubs is shown in Figure 3D. After foliar spraying of the shrubs with GA3 at 50 mg/L and 100 mg/L, no significant difference was observed in the treated plants compared with Ck. However, at 150 mg/L and 200 mg/L, a significant increase was found compared to Ck. After regrowth, plants treated with 50 mg/L, 100 mg/L, and 200 mg/L GA3 had higher contents of PRO (42.68%, 28.80%, and 88.09%, respectively) compared with Ck. In contrast, the PRO content in shrubs treated with GA3 at 150 mg/L decreased by 14.43% compared with Ck. Nevertheless, this decrease was insignificant. After treatment and regrowth of the shrubs, the treatment of 200 mg/L resulted in the highest amount of PRO compared to the other treatments (Ck, 50 mg/L, 100 mg/L, and 150 mg/L).

### 2.8. Effect of GA3 on Antioxidant Enzyme Activity in N. tangutorum

Regarding the superoxidase dismutase (SOD) activity, the results in Figure 3D showed significant differences in *N. tangutorum* shrubs treated with GA3. The SOD activity of shrubs treated with 50 mg/L, 150 mg/L, and 200 mg/L of GA3 increased significantly compared with Ck by 1.21 times, 1.50 times, and 1.74 times, respectively. After regrowth, the SOD activity increased significantly in 50 mg/L, 100 mg/L, 150 mg/L and 200 mg/L GA3 treated shrubs by 1.17 times, 1.30 times, 1.48 times, and 1.27 times, respectively. The results also revealed GA3 treatment at 150 mg/L as one with the dominant effect on SOD activity after regrowth compared with other treatments (Figure 3D).

As regards peroxidase (POD), the activity significantly decreased in shrubs treated with 50 mg/L by 36.81% compared with Ck, while at 200 mg/L, a significant increase of 48.94% was observed. After regrowth, the concentrations of 100 mg/L, 150 mg/L, and 200 mg/L had no significant effect on the POD activity compared with Ck; however, at 50 mg/L, the POD activity significantly increased compared with the Ck by 46.62% (Figure 3E). Additionally, this increase in POD activity at 50 mg/L in the plants after regrowth was also the highest compared to other concentrations and Ck (Figure 3E).

### 2.9. Effect of GA3 on Malondialdehyde Content in N. tangutorum

The malondialdehyde (MDA) content of the shrubs was not significantly enhanced after applying GA3 at all concentrations compared to the control (Figure 3F). In particular, GA3 at concentrations of 50 significantly decreased the content of MDA by almost 0.21-fold compared with Ck. Furthermore, the MDA contents of shrubs treated with concentrations of 150 mg/L and 200 mg/L prior to coppicing increased compared with that of shrubs examined after coppicing by 0.39 times and 0.41 times, respectively. Nevertheless, the contents were still lower than that of Ck. Among the treatments, the application of GA3 at 50 mg/L caused the lowest content of MDA in the shrubs after regrowth by 77.89% compared to Ck.

Overall, the effects of each plant growth regulator (IAA, ABA, and GA3) at different concentrations on these osmolytes and antioxidant enzyme activity were different in *N. tangutorum* after foliar sprays and before and after coppicing. All results showed that concentration was the dominant factor in improving osmolytes and antioxidant enzyme activity in *N. tangutorum*. Compared to the results before and after foliar application, the results after regrowth of *N. tangutorum* shrubs showed more osmolytes (SS, SP, and PRO) and antioxidant enzyme activity (SOD and POD) at all concentrations. Notably, higher concentrations (>50 mg/L) of IAA, ABA, and GA3 significantly increased these parameters in *N. tangutorum* shrubs.

## 3. Discussion

Plant growth regulators (PGRs) are substances that support plant development and improve the stress tolerance of plants under various stress conditions [4,5]. Phytohormones such as auxins control much of the growth and developmental processes of vascular plants [43]. Zhu et al. [44] and Khan et al. [45] also reported that IAA is widely used as a plant growth regulator to promote plant growth. Studies in recent years have shown that exogenous indoleacetic acid (IAA) can also increase the activities of antioxidant enzymes, which can help plants become more resistant and adaptable [46,47,48] Abscisic acid (ABA) has many vital functions in plants at both physiological and molecular levels, especially under environmental stress [19,20]. As a messenger in response to abiotic stress, this phytohormone also promotes the expression of genes encoding antioxidant enzymes and enhances their activity [21,22]. In addition, ABA has a signaling function in controlling of stomata, plant development, and metabolic processes [49]. Gibberellic acid (GA3) protects plants from environmental challenges by controlling the activity of antioxidant enzymes and lowering the excessive levels of intracellular ROS under stressful conditions [26]. Sharma et al. [32] have shown that increased proline contributes to maintaining membrane integrity by reducing lipid oxidation (caused by ROS) and safeguarding the cellular redox potential. The role of proline as a signaling agent in regulating mitochondrial function influences cell proliferation by activating specific genes required for stress recovery. The antioxidant system of plants consists of both enzymatic and nonenzymatic antioxidants that support the ability of plants to survive under pressure [33]. Superoxide dismutase (SOD) is an important protective enzyme in plants [50]. Peroxidase (POD) plays an important role in a number of metabolic processes and plant defense [51]. Malondialdehyde (MDA) is a commonly used and reliable marker for assessing the damage to a stressed plant, as it is one of the by-products of peroxidation of polyunsaturated fatty acids in cells [52].

In this work, the effect of IAA application on osmolyte activities was investigated. Soluble sugar support plant immunity by promoting plant defense responses and metabolism [53]. They also function as signaling molecules that recognize environmental stress and trigger defense responses by controlling the expression of genes and proteins and the assembly of metabolites [54,55]. After IAA treatment and regrowth, SS content increased at 150 mg/L versus Ck. In support of our results, an increase in soluble sugar content by IAA application was also reported in wheat leaves and sheaths [56]. Increasing the amount of soluble sugars can increase the cold tolerance of a plant [57]. The SS content decreased at 150 mg/L and 200 mg/L concentrations after regrowth, suggesting that coppicing with higher IAA concentration could cause the decrease. SP is an important osmoregulatory substance in plants, which can maintain constant cell osmotic pressure and contribute to oxidase activation [58]. Various signaling processes and plant protection under stress conditions are significantly influenced by SP [59]. Increased levels of protein that support the maintenance of a fully acclimated state or stimulate cell division were found in tolerant plants that could adapt to stressful conditions, indicating restoration of plant growth and development [60]. Our results suggest that the SP content in *N. tangutorum* shrubs was higher after treatment with 200 mg/L IAA and after regrowth than in the control. This may be because IAA can control genes through proteins and specific transcription factors that are tuned to environmental responses [61]. Increased soluble protein content by IAA treatment in *Pisum sativum* was also reported by Aldesuquy et al. [62]. According to studies, PRO can act as an osmolyte but is also considered an effective ROS scavenger, metal chelator, protein stabilizer, and inhibitor of programmed cell death [63]. In addition, the IAA application in this study also increased PRO content at 200 mg/L after treatment and after regrowth. Similar results were also observed by Liang et al. [64] in lettuce. Studies in recent years have shown that exogenous IAA application may also help plants become more resilient and adaptable due to its potential to increase the activity of antioxidant enzymes. In this study, IAA treatment before coppicing increased the activities of SOD and POD. After regrowth, it was observed that IAA treatment at 150 mg/L increased the activities of SOD and POD more in *N. tangutorum* shrubs than before. It could be that IAA signal transduction induced the expression of genes related to antioxidant enzymes. It is known that increased activity of antioxidant enzymes in response to environmental conditions reduces the severity of oxidative stress-induced damage in plants [65]. Our results are consistent with other studies reporting increased SOD and POD activities. In particular, Yunmin et al. [66] reported that, at concentrations of 30, 60, and 90 mg/L, SOD and POD activities in *C. betacea* seedlings increased. Khalid et al. [67] also reported an increase in SOD and POD activity by exogenous application of IAA in Chinese cabbage seedlings. MDA content decreased when treated with IAA compared to Ck and increased slightly after regrowth at most concentrations compared to the application of IAA before regrowth, but the difference in the increase was not significant. Reduction in MDA by IAA application was also reported by Ben et al. [68] in pea seedlings, whereas Gong et al. [69] reported similar results in spinach seedlings. Therefore, an appropriate concentration of IAA may increase the osmolyte concentration and antioxidant enzyme activity in *N. tangutorum*, thus improving its adaptability to the environment.

Abscisic acid (ABA) is the primary controller of abiotic stress resistance in plants and coordinates a number of functions [17,18,19,20]. The biosynthetic pathways of osmolytes at the molecular level are controlled by ABA, which acts as a signaling molecule [70,71]. The major osmotic regulators in plants are soluble sugar, free proline, and soluble protein [72]. In the present study, the maximum increases in the accumulation of osmotic regulating substances SS, SP, and PRO were observed in *N. tangutorum* under the influence of ABA after foliar application (200 mg/L and 50 mg/L for SS, 100 mg/L for SP, and 200 mg/L for PRO) and after the regrowth of the shrubs. The increased levels of these osmoprotective substances by ABA stimulation may further reduce oxidative stress [73]. These results are similar to that of Hou et al. [74], where, in their case, ABA was used under drought conditions in *R. soongorica*. Additionally, the application of ABA at 6 mg/L was the only treatment that increased the osmotic regulatory substances, which alleviated the damage of drought stress for the maintenance of *R. soongorica* population in desert areas [74]. Fang et al. [75] also reported that, under drought stress, the accumulation of osmotic regulatory substances such as PRO, SS, and SP in *R. soongorica* increased with the application of ABA. The increase in osmotic regulatory substances by ABA treatment was also observed by Yang et al. [76] in Sabina seedlings under low-temperature stress. One of the most important methods to eliminate ROS is the antioxidant enzyme system in plants [77]. Many studies have shown that ABA can increase the activity of antioxidant enzymes and mitigate stress-induced oxidative damage to plants [78,79,80]. However, the data supporting this hypothesis are contradictory. For example, the application of ABA to *Cotinus coggygria* shrubs under drought stress significantly improved the activity of SOD and POD [81]. It was found that a moderate dose of ABA increased enzymatic activities, suppressed the production of endogenous hormones, and upregulated catabolic genes to improve the survival of rice [82,83]. The current study demonstrated that ABA increased antioxidant enzyme activity, including SOD and POD, after treatment with ABA (at 200 mg/L and 50 mg/L for SOD; 200 mg/L for POD) and after regrowth compared with CK. One possible reason is that the generation of ABA-induced H_2_O_2_ may have triggered the response of the entire antioxidant defense system against oxidative stress. Such enhancement of antioxidant defenses is capable of scavenging increased H_2_O_2_ concentrations. MDA levels were lower in the treatments of ABA than in CK, but increased slightly after regrowth, although the differences were not significant. Similar results were also reported by Hou et al. [74] and Zhang et al. [77]. In this study, no stress was combined with treatments, but it was observed that, even in the absence of stress, the application of ABA enhanced the osmotic adaptation substances of *N. tangutorum*, which can effectively regulate their metabolism and resistance to abiotic stress. In addition, ABA stimulated the activity of antioxidant enzymes, eliminating the overaccumulation of ROS by reducing lipid peroxidation and protecting the cell membrane from oxidative damage.

It has been reported that GA3 is an important phytohormone that promotes phys-iological and developmental processes in plants, such as root formation, flowering, and maturity, as well as seed germination, cell division, and root growth [84]. GA3 plays a critical role in plant defense by reducing intracellular ROS formation and triggering the defense system [26,67,84,85,86]. Osmotic regulation is one of the most popular methods for maintaining cell turgor when tissue water potential decreases in plants [87]. It involves the production and accumulation of compatible organic solutes, including those that are normally nontoxic at high cytosolic levels, such as SS, SP, and PRO [88,89]. As vital osmotic regulators and stabilizers of the membrane system, free proline and soluble protein also increase the ability of plants to withstand stress [90]. Foliar application of GA3 to maize seedlings reportedly increased SP levels under salt stress [91]. For PRO content, a GA3 treatment of 300 µM under a 100 mM NaCl treatment had the significantly highest values after day 20 and day 30, indicating that SP and PRO can accumulate under salt stress by GA3 treatment. Furthermore, an increase in SP in *Sesbania* pea by GA3 application was reported by Guo et al. [92]. Similar results were also observed by Khalid and Aftab [67]. In our case, GA3 application was not associated with stress, but the current study showed that SP and PRO contents increased after GA3 treatment, with a strong increase observed at GA3 concentration of 200 mg/L for both contents (SS and PRO) after treatment and after regrowth. To mediate plant stress responses, soluble sugars act as signaling molecules that interact with plant hormones [54,55]. Our results showed that spraying *N. tangutorum* with GA3 increased the SS content. These results are supported by Hu et al. [93], who found an improvement in SS content by GA3 in toon sprouts. Similar results on the osmotic function of GA3 in different plant species were reported by previous studies [94,95,96]; however, in these studies, the increase was under oxidative stress. Plant cells have both enzymatic and non-enzymatic antioxidant defense systems [48,88,97]. These enzymatic antioxidants include SOD and POD [48,88,98]. Many organisms have evolved ROS-scavenging systems, which may be enzymatic or nonenzymatic, that allow cells to maintain nontoxic and stable levels of ROS. The enzymatic ROS-scavenging system is composed of superoxide dismutase and peroxidases [99]. In the present study, a significant increase in the activities of SOD and POD was observed in *N. tangutorum* when GA3 was administered and after regrowth, suggesting that an appropriate GA3 treatment enhances the protective enzyme activities of *N. tangutorum* and removes excess ROS. This can ward off some degree of damage and ultimately ensure normal growth of the shrub. These results are in agreement with those of Adam et al. [100], who reported the increase in SOD and POD by GA3 application in sorghum seedlings. Another study also mentioned that GA3 treatment significantly increased the activity of POD and SOD in castor bean seedlings treated with 200 and 250 µM GA3, respectively [101]. MDA is a byproduct of lipid peroxidation of membranes, which may reflect the degree of cell membrane damage [102]. The increased antioxidant activity can be interpreted as an indication of decreased accumulation of MDA content [103,104,105]. MDA content decreased after GA3 treatment compared with CK but increased slightly after regrowth than when treatment was applied before coppicing. An increase in MDA content indicates that electrolyte losses may occur, resulting in loss of membrane integrity [106]. In our case, it is possible that coppicing had a stress effect on the plant, which explains this finding. The enhancement of antioxidants and osmolytes by GA3 is an effective mechanism to increase the tolerance of *N. tangutorum* to oxidative stress.

The findings of this study are based on the effects of each plant hormone applied at different concentrations. Thus, the impact of the three PGRs on *N. tangutorum* was not compared and is recognized as a study limitation.

## 4. Materials and Methods

### 4.1. Experimental Site

The experimental site was in Wuwei district. Wuwei district is located between latitudes 37°23′ and 38°12′ N and longitudes 101°59′ and 103°23′ E in the northwest of Lanzhou, Gansu Province, China. It is also the eastern end of the Hexi Corridor and lies north of the Qilian Mountains. The total area is 5.08 × 10^5^ hm^2^, of which 1.28 × 10^5^ hm^2^ is arable land. The soil in Gansu is a calcareous desert soil (sandy clay loam, typical anthrosol), which is classified as “irrigated desert soil” according to the Chinese Soil Classification System [107] and corresponds to an Anthropic Camborthid soil according to the “Keys to Soil Taxonomy” [108]. Carbonate contents in the soils of the study area range from 4.5% to 14.6%. The parent material consists of alluvial deposits, pedogenic rocks, including carbonatite, fine metamorphic rocks, and marine volcanic rocks.

### 4.2. Plant Material

Shrubs of *N. tangutorum* were sampled from Wuwei, Gansu Province, China and used for this study (Figure 4). The shrubs were selected at the maturation stage.

### 4.3. Hormones Preparation

Briefly, the desired weight (0.5 g) of IAA, ABA, and GA3 powder was dissolved in (1.5 mL of 50% ethanol) and diluted to 100 mL of distilled water in a 1 L standard flask. The solution was then made up to the 1000 mL mark with distilled water to obtain a final stock solution. This final stock solution was then serially diluted to obtain four different concentrations of IAA, ABA, and GA3: 50 mg/L, 100mg/L, 150mg/L, and 200mg/L.

### 4.4. Experimental Design

The experiment was organized in a completely randomized design (CRD) with three replicates. A total of 150 *N. tangutorum* shrubs were selected for data collection at maturity on 11 June 2021. The leaves of the selected shrubs were also removed for data collection. Then, IAA, ABA, and GA3 were applied as foliar spray at different concentrations of 50 mg/L, 100 mg/L, 150 mg/L, and 200 mg/L on the plants that served as the treatment group. After 2 days (13 June 2021), a second leaf sample was collected for data collection and the shrubs were cut (coppiced). Four months later (4 October 2021), after the shrubs had regenerated, a third leaf sample was taken for data collection (soluble sugar, soluble protein, proline, and antioxidant enzyme activity content). A control group was formed with shrubs treated only with distilled water and not with IAA, ABA, or GA3. All leaves sampled before and after the application of IAA, ABA, and GA3, and after the regrowth of the shrubs were placed in liquid nitrogen for preservation and brought to the laboratory for biochemical analysis. Thus, the analyses were performed on the leaves collected before and after treatment with IAA, ABA, GA3, and distilled water, after coppicing and regeneration of the shrubs.

### 4.5. Analyzes

#### 4.5.1. Determination of Soluble Sugar (SS)

Soluble sugar content was determined using the colorimetric method described by Zhang et al. [109] with slight modifications. Briefly, 0.2 g of samples and 8 mL of distilled water were placed in test tubes, which were placed in a boiling water bath for 30 min and allowed to cool. Then, 0.5 mL of ethyl acetate ketone and 5 mL of sulfuric acid were mixed, placed in a boiling water bath for 10 min, and cooled. The light absorbance value of each sample was measured at 630 nm using a Genesis 10S UV/Vis spectrophotometer (Thermo Fisher Scientific, Waltham, MA, USA). Sugar content was determined using a standard linear equation and then calculated in mg/g.

#### 4.5.2. Determination of Soluble Proteins (SP)

Soluble protein content was determined with slight modification to the method described by Zhu et al. [110] using Coomassie brilliant blue. First, 0.5 g of the leaves of *N. tangutorum* were ground to a homogenate with 2 mL of phosphate buffer. The supernatant was centrifuged at 12,000 r/min for 20 min at 4 °C. Then, 1 mL of the supernatant, 1 mL of water, and 5 mL of Coomassie Bright blue g-250 were added, followed by centrifugation and subsequent sample oscillation. The light absorbance value was measured at 595 nm using a Genesis 10S UV/Vis spectrophotometer (Thermo Fisher Scientific, Waltham, MA, USA). The same procedure was used to prepare a control using 5 mL of Cowmas bright blue g-250 solution and distilled water. The absorbance was then determined by spectral measurements to determine the protein content using a standard curve.

#### 4.5.3. Determination of Proline (PRO)

Proline content was assayed according to Yousaf et al. [111] method with slight modifications. Fresh leaves of *N. tangutorum* (0.5 g) were ground in 2 mL of 3% (*w*/*v*) sul-fosalicylic acid and boiled in a water bath at 90 °C for 10 min to extract the proline. Then the mixture was rapidly cooled to room temperature and centrifuged at 3000 rpm for 10 min. Then, 2 mL of freshly prepared acid–ninhydrin solution and 2 mL of glacial acetic acid were added to the tubes containing 2 mL of supernatant. The tubes were heated in a boiling water bath for 30 min. The reaction mixture was then extracted with 4 mL of toluene at room temperature and kept away from light for 4 hrs. The absorbance at 520 nm was determined spectrophotometrically using toluene as a blank.

#### 4.5.4. Determination of Antioxidant Enzyme Activity 

Using a pestle and mortar, leaves of *N. tangutorum* (0.5 g; *n* = 6) were homogenized in 6 mL of 0.05 mol/L ice-cold potassium phosphate buffer (pH 7.8). The homogenate was placed in a polypropylene centrifuge tube and centrifuged at 10,000× *g* for 15 min. The supernatant was used to measure the activity of antioxidant enzymes. All procedures were performed at 4 °C. The activity of superoxide dismutase (SOD) and peroxidase (POD) was measured according to the method of Illescas et al. [112].

#### 4.5.5. Determination of Malondialdehyde (MDA) Content

The analysis of MDA in leaves was performed according to the method of Todorova et al. [113] with slight modifications. This method is based on the reaction with thiobarbituric acid. Fresh leaves (0.2 g) were ground homogeneously in 20 mL of 10% trichloroacetic acid solution and centrifuged at 3000 rpm for 10 min. One milliliter of the supernatant was mixed with 2 mL of 20% TCA solution containing 0.6% thiobarbituric acid. The mixture was heated at 95 °C in a water bath for 15 min, cooled, and centrifuged at 3000 rpm for 10 min. The absorbance of the supernatant was measured at 450 nm, 532 nm, and 600 nm.

### 4.6. Statistical Analysis

All the data were entered into excel and the significance level among means was analyzed using Duncan’s multiple-range tests (*p* ≤ 0.05) after one-way ANOVA analysis in SPSS statistical software (Version 22.0, SPSS Inc., Chicago, IL, USA). All graphical presentations were performed using GraphPad Prism 8 (GraphPad Software Inc, La Jolla, CA, USA).

## 5. Conclusions and Recommendations

This study examined the effects of PGRs IAA, ABA, and GA3 on osmotic regulatory substances and antioxidant enzyme activity in *N. tangutorum* after foliar sprays and before and after coppicing. The results indicated that each PGR at different concentrations caused variations in osmotic regulatory substances and antioxidant enzyme activity in *N. tangutorum* after treatment and before and after coppicing. Our results indicate that all PGRs increased osmotic regulatory substances (SS, SP, PRO) and antioxidant enzymes (SOD, POD) activity in *N. tangutorum*, especially at concentrations above 50 mg/L. In addition, the PGRs led to a significant decrease in MDA content in *N. tangutorum*. As a result, *N. tangutorum* shrubs became resistant to various biotic stresses, which improved their adaptability to the environment.

Considering the limitations and taking advantage of the potential benefits of this study to determine its application in forest biotechnology, these results need to be extended in future studies by comparing the effects of these three PGRs with or without stress (drought stress, salt stress, etc.) at higher concentrations (above 50 mg/L) on the antioxidants, osmolytes, genes, and other physical aspects of this shrub in a single study. This will provide a thorough knowledge of the functioning of *N. tangutorum* shrubs and better ways to propagate them to combat desertification.

## Figures and Tables

**Figure 1 plants-11-02559-f001:**
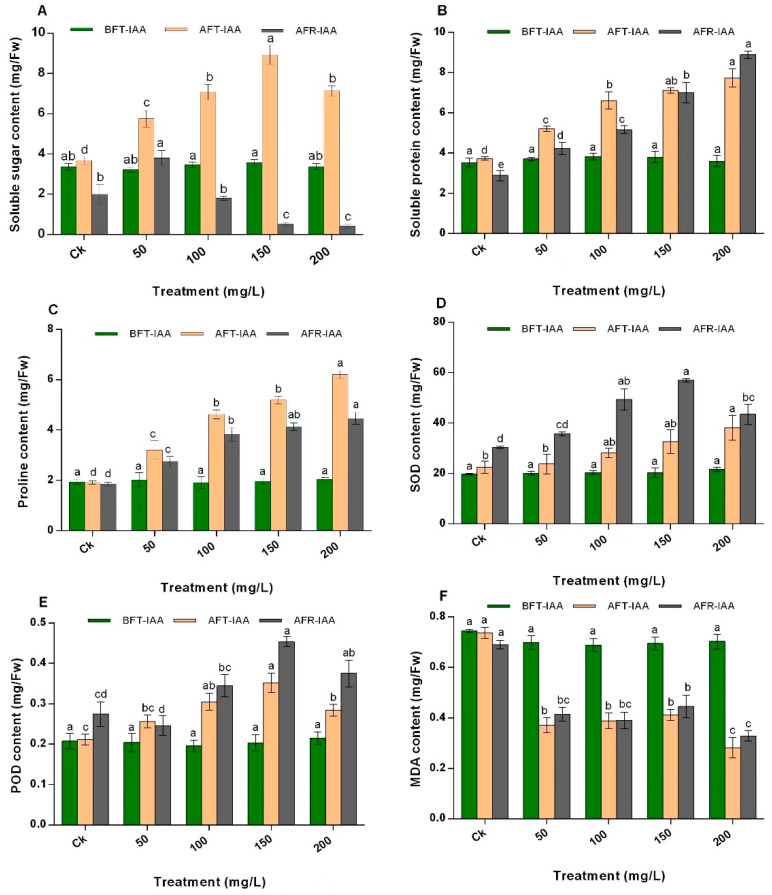
Effect of IAA on the (**A**) soluble sugar content, (**B**) soluble protein content, (**C**) proline content, (**D**) SOD activity, (**E**) POD content, and (**F**) MDA content of *N. tangutorum*. The values represent the mean of three replicates determinations. The error bars represent standard deviations. Bars with the same letter are not significantly (*p* > 0.05) different.

**Figure 2 plants-11-02559-f002:**
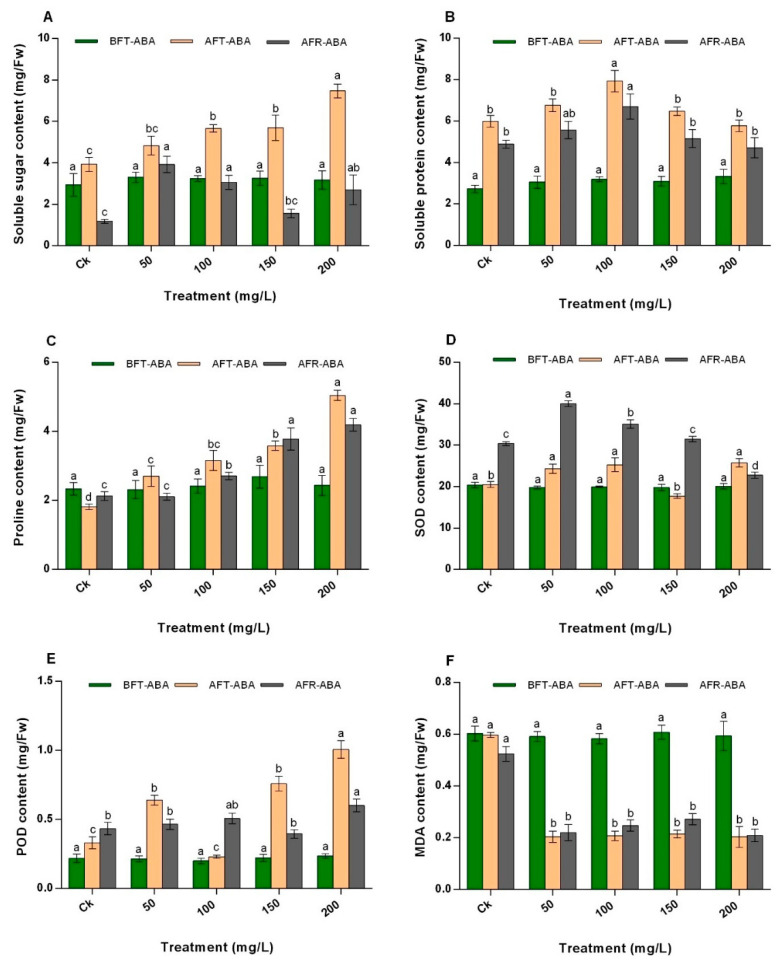
Effect of ABA on the (**A**) soluble sugar content, (**B**) soluble protein content, (**C**) proline content, (**D**) SOD content, (**E**) POD content, and (**F**) MDA content of *N. tangutorum*. The values represent the mean of three replicates determinations. The error bars represent standard deviations. Bars with the same letter are not significantly (*p* > 0.05) different.

**Figure 3 plants-11-02559-f003:**
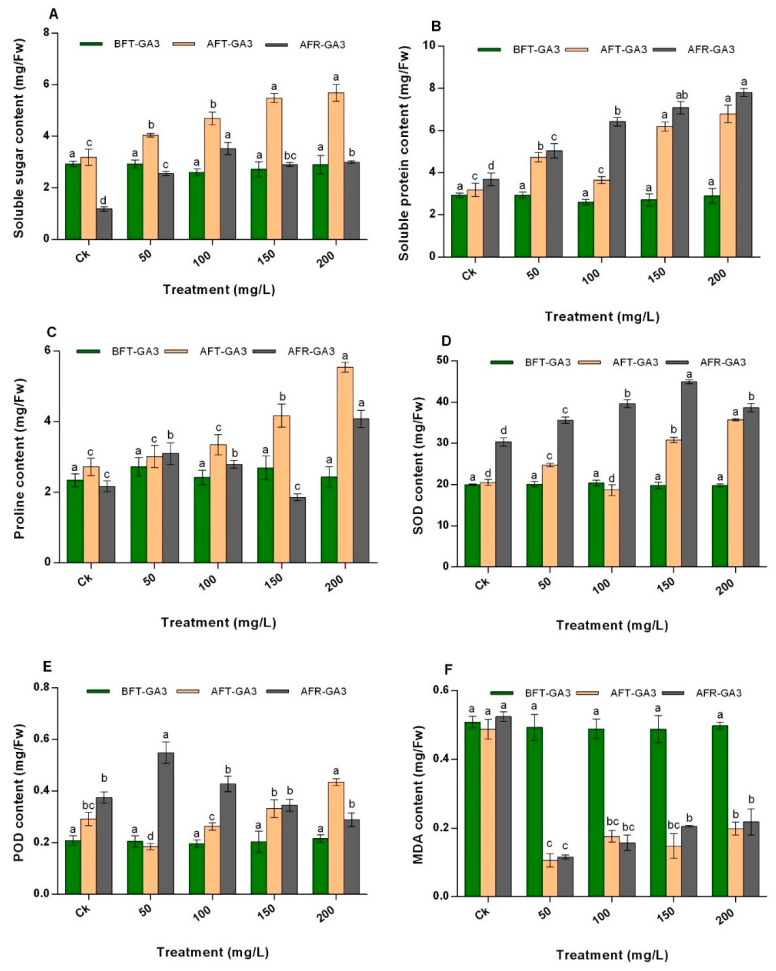
Effect of GA3 on the (**A**) soluble sugar content, (**B**) soluble protein content, (**C**) proline content, (**D**) SOD content, (**E**) POD content, and (**F**) MDA content of *N. tangutorum*. The values represent the mean of three replicates determinations. The error bars represent standard deviations. Bars with the same letter are not significantly (*p* > 0.05) different.

**Figure 4 plants-11-02559-f004:**
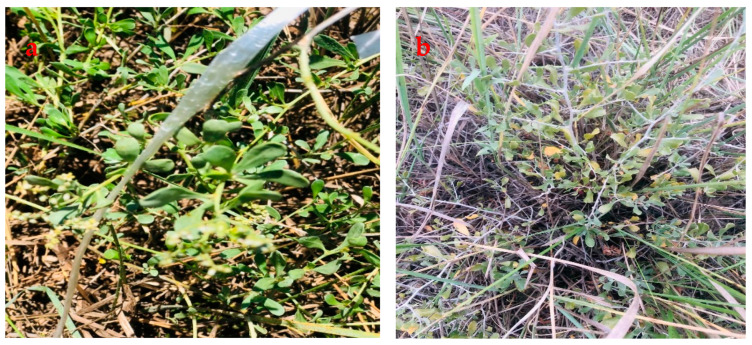
(**a**) Picture of selected *Nitraria tangutorum* shrubs at the maturation stage from the experimental site before coppicing and (**b**) picture of selected *Nitraria tangutorum* shrubs from the experimental site after regrowth.

## Data Availability

The data used in this study are available on request from the corresponding author.

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
