# Peer review of "Effect of Plant Growth Regulators on Osmotic Regulatory Substances and Antioxidant Enzyme Activity of Nitraria tangutorum"

_plants, 2022, doi:10.3390/plants11192559_

Round 1

Reviewer 1 Report

Interesting research presented in the publication, supported by a large amount of scientific literature. Nitraria tangutorum Bobr. is endemic to China. It has extremely important ecological value due to its broad development prospects for the regulation of desertification and ecological environment improvement in the Qinghai-Tibet Plateau. berry plant is relatively rich in nutrients such as amino acids, fatty acids, mineral elements, and a variety of biologically active ingredients such as polysaccharides, flavonoids, alkaloids, and anthocyanins . It has a broad medical and pharmacologic value, being used for the treatment of various diseases and with known antifatigue, antioxidation, hypolipidemic, and hypoglycemic properties, protection of chemical liver damage, and being involved in immune regulation.

what needs to be improved:

to explain the abbreviation  MDA in abstract

check the units given in figure 1

it is a pity that the effects of the plant hormones have not been compared

the photos are not readable, they do not indicate much, it should be better described in the title

please inform what amounts of plant hormone solutions were used in foliar spraying

See References Latin names of plants should always be italicized. The first letter of the genus name is capitalized but the specific epithet is not, e.g. Lavandula angustifolia.

Reviewer 2 Report

Comments for Author
I have read your paper mentioned " 
Effect of plant growth regulators on osmotic regulatory substances and antioxidant enzyme activity of Nitraria tangutorum First of All, English should be improved because there are many grammatical and abbreviation shortness. Finally, this paper could be accepted after major revision. 

Some specific comments are

Most of sentences unclear in the introduction. Please recheck again all sentences.
Author discussed so general and used many old references regarding ROS, POD and other enzymatic activity. . Regarding enzymatic analyses such as antioxidant or ROS
Please support with latest refs.   Please incorporate them in the introduction.

Discussion is so general, I could suggest to the Author more attention. Conclusion is like results sections. It is not good. Author should give their recommendations.

Best Regards

Round 2

Reviewer 2 Report

Accept for publication